# Experimental and Simulation Research on Femtosecond Laser Induced Controllable Morphology of Monocrystalline SiC

**DOI:** 10.3390/mi15050573

**Published:** 2024-04-26

**Authors:** Yang Hua, Zhenduo Zhang, Jiyu Du, Xiaoliang Liang, Wei Zhang, Yukui Cai, Quanjing Wang

**Affiliations:** 1School of Mechanical and Electronic Engineering, Shandong Jianzhu University, Jinan 250101, China; huayang20@sdjzu.edu.cn (Y.H.); zhangzhenduo163@163.com (Z.Z.); 24377@sdjzu.edu.cn (J.D.); zhangweisedu@163.com (W.Z.); 2Department of Industrial and Systems Engineering, The Hong Kong Polytechnic University, Hung Hom, Kowloon, Hong Kong, China; mexiaoliang.liang@polyu.edu.hk; 3School of Mechanical Engineering, Shandong University, Jinan 250061, China

**Keywords:** femtosecond laser, multi-physics model, carrier density, controllable morphology, SiC

## Abstract

Silicon carbide (SiC) is utilized in the automotive, semiconductor, and aerospace industries because of its desirable characteristics. Nevertheless, the traditional machining method induces surface microcracks, low geometrical precision, and severe tool wear due to the intrinsic high brittleness and hardness of SiC. Femtosecond laser processing as a high-precision machining method offers a new approach to SiC processing. However, during the process of femtosecond laser ablation, temperature redistribution and changes in geometrical morphology features are caused by alterations in carrier density. Therefore, the current study presented a multi-physics model that took carrier density alterations into account to more accurately predict the geometrical morphology for femtosecond laser ablating SiC. The transient nonlinear evolutions of the optical and physical characteristics of SiC irradiated by femtosecond laser were analyzed and the influence of laser parameters on the ablation morphology was studied. The femtosecond laser ablation experiments were performed, and the ablated surfaces were subsequently analyzed. The experimental results demonstrate that the proposed model can effectively predict the geometrical morphology. The predicted error of the ablation diameter is within the range from 0.15% to 7.44%. The predicted error of the ablation depth is within the range from 1.72% to 6.94%. This work can offer a new way to control the desired geometrical morphology of SiC in the automotive, semiconductor, and aerospace industries.

## 1. Introduction

Silicon carbide (SiC) has been utilized in the automotive, semiconductor, and aerospace industries due to its appealing features, including a wide bandgap, strong thermal conductivity, and remarkable chemical inertness at extreme temperatures [1,2,3]. Nevertheless, owing to the inherent high hardness and brittleness of SiC, conventional machining techniques have many shortcomings, including surface microcracks and pits, low geometrical precision, and severe tool wear [4]. These shortcomings seriously affect the reliability and fatigue life of the functional components during their service [5]. Consequently, the research of high geometrical precision and high surface quality has become a hot topic in the field of SiC manufacturing and it is of great value to establish a predictive model. Femtosecond lasers have the characteristics of ultrashort pulse duration and strong peak intensity, which can effectively inhibit heat-induced cracks, pits, and residual stress. It has been demonstrated that the heat-affected zone of materials is suppressed and that the surface integrity can be greatly enhanced when the pulse width decreases from nanoseconds to femtoseconds [6,7,8]. Consequently, femtosecond lasers are utilized on materials that are difficult to process, frequently as a way to generate innovative sensors that may be used in harsh environments [9,10,11].

The geometrical morphology of SiC ablation is related to complex factors such as laser intensity, pulse width, and scanning speed and times. Chen et al. [12] carried out the femtosecond laser ablation experiments to investigate the influence of femtosecond laser parameters on geometrical characteristics. They concluded that the greater laser power and smaller pulse width induced the larger ablation depth of SiC. Zhang et al. [13] performed femtosecond laser ablation experiments to analyze the processing parameters impact on the grooves. The experimental results demonstrated that as the power and pulse repetition rate increased and the scanning velocity decreased, there was a corresponding increase in both the depth and width of the groove. The investigation on the effects of femtosecond laser parameters on the oxidation phenomena was studied by Zhai et al. [14]; the experimental results proved that the oxidation phenomenon got more noticeable as the laser power, repetition frequency, and scanning times increased. Additionally, as the scanning velocity increased, the oxidation phenomenon progressively diminished. Xie et al. [15] discovered that the laser ablation phenomenon began to occur when the laser energy density reached the value of 0.33 J/cm^2^ during the single-pulse laser irradiation; no periodic rippling structures appeared on the SiC surface. Zhang et al. [16] found that when the laser energy density declined, the polished surface roughness increased and the laser ablation capacity and polish depth reduced. Hu et al. [17] reported that different types of surface micro-nano structure were produced by femtosecond lasers with distinct pulse overlaps.

Even though the experimental method is accurate in practice, high costs are required in each case. Rather than being based on experiments, the analytical approaches are rooted in physics, which can better reflect actual physical phenomena with geometrical characteristics. Ren et al. [18] proposed a semi-classical two-temperature model for predicting lower electron temperature, lattice temperature, and phase transition. A visualization model based on the two-temperature equation was developed by Chen et al. [19] to simulate the ablation morphology of silicon under single-pulse femtosecond laser processing. Lin et al. [20] established a two-temperature model with the consideration of material optical and thermal-physical properties that were dependent on electron temperature to illustrate the surface melting phenomenon. Wang et al. [21] proposed a hybrid approach that combined molecular dynamics and the two-temperature model, during which the femtosecond laser processing was simulated and the ablation mechanism of material was revealed. A two-temperature model with the consideration of molecular dynamics was established by Wu et al. [22] to study the evolution of temperature, density, and pressure during the ablation of a single-pulsed laser. Zhang et al. [23] used a two-temperature model in conjunction with a hydrodynamics model to simulate the melting and ablation process of microstructure formation on material surfaces irradiated by femtosecond laser. Wang et al. [24] developed a finite element model to investigate the effect of laser energy on the ablation depth, concluding that when the single pulse energy exceeded 8 J/cm^2^, the predicted ablation depth was essentially unaffected by the incident laser pulse energy. An et al. [25] put forward a two-temperature model that combined with a novel molecular dynamics method to predict the ablation morphology of femtosecond laser. Zhai et al. [26] adopted the theoretical calculations and wave optics simulations to analyze the results of SiC by femtosecond laser processing. The simulation results could aid in explaining the causes of various ablation morphologies.

In accordance with the aforementioned subject, although there are many prediction models of femtosecond laser ablation, few studies have taken into account the carrier density alterations. Furthermore, the transient nonlinear evolutions of the optical and physical characteristics of femtosecond lasers ablating SiC cannot be analyzed by existing models and experiments because of the extremely short timescale of the femtosecond laser irradiation process. This work proposed a multi-physics model that considered carrier density alterations to more accurately predict the geometrical morphology of SiC generated by femtosecond laser irradiation. At the same time, the model illustrated the transient nonlinear evolutions of optical and physical characteristics during the ablation of SiC. A theoretical foundation for the investigation of laser-induced controlled morphology was provided with the establishment of the multi-physics field model.

To validate the forecasting ability of the model, relevant experiments were carried out in which monocrystalline SiC was ablated using various femtosecond laser powers. The model of the temperature field was established by taking the carrier density alterations into account based on COMSOL^®^ Multiphysics 6.0. The femtosecond laser ablation experiments were performed and the ablated surfaces were subsequently analyzed by means of scanning electron microscopy (SEM), confocal laser scanning microscopy (CLSM), and energy dispersive spectroscopy (EDS). The experimental results were compared with the predicted results to prove the validity of the proposed model. Moreover, the influence of laser parameters on the ablation morphology and the effect of carrier density alterations on the temperature field were analyzed. The required ablation morphology might be processed by choosing the appropriate laser parameters in this study. The present work could provide new insights into guiding the selection of femtosecond laser processing conditions in the automotive, semiconductor, and aerospace industries.

## 2. Theoretical Model

### 2.1. Principle Description

The physical process of femtosecond laser interaction with SiC includes two stages: (i) the laser energy is first absorbed by the electron system via photon-electron interaction and then (ii) the electron energy is transmitted to the lattice subsystem via electron-lattice energy coupling within a few picoseconds. In contrast to the ablation process of traditional pulsed lasers, the ablation process of femtosecond laser could not be directly represented by a macroscopic heat exchange model. As the two-temperature model could describe the energy transmission in photons, electrons, and lattice, it was employed to represent the physical process of femtosecond laser interaction with SiC [27,28,29]:(1)$$C_{e}\frac{\partial }{\partial t}T_{e}=K_{e}\frac{\partial^{2}T_{e}}{\partial z^{2}}-g(T_{e}-T_{l})+Q(z,r,t)$$
(2)$$C_{l}\frac{\partial }{\partial t}T_{l}=K_{l}\frac{\partial^{2}T_{e}}{\partial z^{2}}+g(T_{e}-T_{l})$$
where *Q*(*z*,*r*,*t*) stood for the light source term, *z* represented the laser irradiation direction that is perpendicular to the surface of the material, *t* represented the action time, and *r* represented the radial coordinate from the center of laser beam to its edge. *T_e_* and *T_l_* denoted the electron temperature and the lattice temperature, respectively, *K_e_* and *K_l_* denoted the electron thermal conductivity and the lattice thermal conductivity, respectively, *C_e_* and *C_l_* stood for the heat capacity of electron and lattice, respectively. *g* was the electron-lattice coupling coefficient.

In general, femtosecond lasers produce Gaussian light that can be considered a point source of light when they are intensely focused. From the standpoint of laser energy, this light can also be roughly described as a point source of heat. When the femtosecond laser is irradiated on the surface of SiC, the function of laser intensity on SiC surface could be obtained by the follow equation [30,31]:(3)I(z=0,r,t)=2Fπln2τp(1−R)exp−4ln2t−ln1002ln2τpτp2exp−r2r02
where *F* and *R* were the laser energy density and the material reflectivity, respectively. *τ_p_* and *r*_0_ were the pulse width and the waist radius of laser beam, respectively.

When the electrons were stimulated to ionize through linear and nonlinear absorption, the laser power density in the material decayed along the incident direction. Therefore, the laser intensity *I*(*z*,*r*,*t*) at the depth of *z* could be given by the following equation [32,33]:(4)$$I\left(z,r,t\right)=\frac{\frac{\alpha_{1}+\alpha_{FCA}}{2\alpha_{2}}I\left(z=0,r,t\right)\exp\left(z\left(\alpha_{1}+\alpha_{FCA}\right)\right)}{\frac{\alpha_{1}+\alpha_{FCA}}{2\alpha_{2}}+\left(1-\exp\left(z\left(\alpha_{1}+\alpha_{FCA}\right)\right)\right)I\left(z=0,r,t\right)}$$
where *α_FCA_* represented the absorption coefficient of carrier, and *α*_1_, *α*_2_, *α*_3_ represented the absorption coefficient of single-photon, two-photon, and three-photon, respectively.

### 2.2. Carrier Density and Laser Source

During the interaction between the femtosecond laser and SiC materials, the carrier density varied with electron temperature. In the conventional two-temperature model, however, the carrier density remained constant. Therefore, to more truly describe the physical interaction process between femtosecond laser and SiC materials, a new model was proposed that took into account carrier density alteration. It was possible to ignore the impact of electron diffusion on the alterations in carrier density because of the extremely short time scale of the femtosecond laser pulse. The following partial differential equation described the development of carrier density with time [34,35].
(5)$$\frac{\partial n_{e}}{\partial t}=\frac{\alpha_{1}I\left(z,r,t\right)}{hv}+\frac{\alpha_{2}I{\left(z,r,t\right)}^{2}}{2hv}+\frac{\alpha_{3}I{\left(z,r,t\right)}^{3}}{3hv}+\beta n_{e}-\gamma n_{e}$$
(6)$$n_{e}=\frac{\left(1-R\right)I\left(z,r,t\right)\tau_{p}}{1.763hv}\left[\alpha_{1}+\frac{\left(1-R\right)\alpha_{2}I\left(z,r,t\right)}{3}\right]$$
where *h* represented the Planck constant, *v* was the frequency of the light wave, *β* stood for the collisional ionization coefficient, *γ* was the Auger recombination coefficient, and *n_e_* represented the carrier density.

The total source term *Q*(*z*,*r*,*t*) was represented by the following [36]:(7)$$Q\left(z,r,t\right)=(\alpha_{1}+\alpha_{2}I\left(z,r,t\right)+\alpha_{3}I{\left(z,r,t\right)}^{2}+\alpha_{FCA})I\left(z,r,t\right)-\left(E_{g}+3K_{B}T_{e}\right)\frac{\partial n_{e}}{\partial t}$$
where *E_g_* represented the material bandgap and *K_B_* was the Boltzmann constant.

### 2.3. Boundary Conditions

It was imperative to establish the boundary conditions within the model in order to constrain the thermal transmission relationship between the heat conduction system and the surrounding environment. Under the assumption that the untreated surfaces of the SiC were in an adiabatic state, the electron system received the input heat from the laser heat source, while the lattice system acquired energy through the electron-lattice energy coupling. As a result, the following boundary conditions were established [37]:(8)$$-\lambda \left(\frac{\partial T_{e,l}}{\partial \tau_{p}}\right){|}_{r=l}=0$$
(9)$$-\lambda \left(\frac{\partial T_{e,l}}{\partial \tau_{p}}\right){|}_{z=-h}=0$$
(10)$$-\lambda \left(\frac{\partial T_{e}}{\partial \tau_{p}}\right){|}_{z=0}=Q$$
(11)$$-\lambda \left(\frac{\partial T_{l}}{\partial \tau_{p}}\right){|}_{z=0}=g\left(T_{e}-T_{l}\right)$$
where *λ* was the thermal conductivity, and *l* and *h* were the length and height of the axisymmetric model, respectively.

### 2.4. Modeling Process

COMSOL^®^ Multiphysics was used to create a model that took carrier density alterations into account. The quantity of laser energy absorbed by the material during the extremely brief femtosecond laser ablation period was significantly more than the amount lost to convection and thermal radiation. Consequently, the simulation only needs to concentrate on the thermal transfer mechanisms occurring within the material internal lattice and electron systems. The differential Equations (1) and (2) would then be used to provide a description of temperature changes in the electrons and lattice, respectively. Figure 1 shows the modeling framework for the geometrical properties.

## 3. Experimental Procedures

The workpiece, which measured 10 mm × 10 mm × 0.3 mm, was made of the material 4H-SiC (Hefei Kejing Materials Technology Co., Ltd., Hefei, China) in the current investigation. As seen in Figure 2a, the femtosecond laser micromachining equipment was used for the ablation tests. An HR-Femto50 femtosecond laser (Wuhan Huaray Precision Laser Co., Ltd., Wuhan, China) with a wavelength of 1030 nm, which has a spot diameter of 20 μm and a pulse width and a maximum power of 300 fs and 7.67 W, respectively. The 4H-SiC material was positioned on a platform with three directions of linear motion and the material surface was subjected to a laser beam for processing.

The femtosecond laser processing schematic diagram is shown in Figure 2b. It included an acousto-optic modulator (AOM) and additional parts made up a laser source that could produce a femtosecond laser beam. After that, a beam expander (BE) was used to enlarge the laser beam and a computer was used to regulate the various parameters of the laser source. Three mirrors (M1, M2, and M3) were used to modify the laser beam direction before it was sent via a non-polarizing beam splitter (BS). Lastly, an objective lens focused the femtosecond laser on the workpiece and a CCD camera recorded the entire ablation procedure.

To study the quantitative relationship between the ablation morphology and the laser power, many sets of tests must be conducted. Based on the maximum power (7.67 W) of the experimental setup, specific laser powers (2.64 W, 3.52 W, 4.35 W, 5.09 W, 5.74 W, 6.28 W, 6.83 W, 7.25 W, and 7.67 W) were chosen. One-to-one correspondence between laser power and laser fluence (4.14 J/cm^2^, 5.57 J/cm^2^, 6.92 J/cm^2^, 8.10 J/cm^2^, 9.14 J/cm^2^, 9.99 J/cm^2^, 10.82 J/cm^2^, 11.54 J/cm^2^, and 12.21 J/cm^2^, respectively) was established, and the laser fluence was the fluence at the center of the Gaussian beam. Table 1 displayed the laser parameter settings. The RCA cleaning procedure was used to remove contaminants prior to femtosecond laser ablation. For each laser power, at least three ablation experiments were carried out in order to decrease experimental error. Using confocal laser scanning microscopy (VK-X260K, Keyence, Osaka, Japan), the ablation diameters and depths were measured. Using a scanning electron microscope (Gemini SEM 300, Carl Zeiss AG, Oberkochen, Germany), surface geometrical morphology was observed.

## 4. Results and Discussion

### 4.1. Ablation Geometrical Characteristics Validation

The ablation requirement would be ascertained by discovering the isotherm that most closely matched the geometrical morphology of the ablation hole, in accordance with the phase-explosion mechanism and the Coulomb explosion mechanism of the femtosecond laser ablation [38,39,40]. In the validation experiment, the ablation requirement of the model was investigated by employing a laser power of 6.83 W. The ablation morphology is seen with CLSM as shown in Figure 3a, where the ablation hole depth is 0.72 μm and the surface machining diameter is 18.2 μm. The isotherm distribution of SiC at various temperatures following 6.83 W laser ablation is displayed in Figure 3b. Figure 3b demonstrates that the lattice system isotherm has a parabolic form and looks like a bowl-shaped pit. As shown in Figure 3c,d, it is observed that the isotherm at 10,976 K is closest to the experimental results when the laser power is 6.83 W. As a result, the 10,976 K isotherm was employed to denote the ablation boundary, and the region that rises above the critical temperature was thought to have met the ablation criterion.

Several simulations based on the proposed model were performed and compared with the experimental results. As indicated in Figure 4a, the laser power is adjusted from 3.52 W to 7.25 W, and the ablation diameter prediction error is regulated between 0.15% and 7.44%. The highest ablation diameter predicted error was 15.44% when the laser power was 2.64 W. As seen in Figure 4b, the laser power is adjusted from 4.35 W to 6.83 W, and the ablation depth prediction error is controlled within the range of 1.72%~6.94%. With a laser power of 2.64 W, the greatest ablation depth prediction error was 54.15%. The experimental and predicted values were mostly consistent, but there was some data fluctuation due to measurement errors. The results demonstrated that the prediction errors in ablation depth and diameter were controlled within 6.94% and 7.44%, respectively.

It was noted that the predicted ablation depths were lower than the experimental results when the laser power increased from 2.64 W to 5.09 W. The predicted ablation depths were greater than the experimental results when the laser power was increased from 5.74 W to 7.67 W. An explanation for the prediction inaccuracy of ablation depth might be the lack of hydrodynamic processes in the model, including the diffusion of molten materials [41]. When the laser power was less than 5.09 W, less molten material was produced during the laser ablation process, which made it simpler to disengage and resulted in a larger actual depth. When the laser power was higher than 5.74 W, laser ablation produced several molten materials that combined to impede ablation, resulting in a predicted depth larger than the actual depth. Furthermore, the energy could not be transported vertically without spreading to adjacent areas after the laser ablation reached a specific depth [42]. The saturation depth of the single-pulse femtosecond laser ablation was not considered in the simulation. As a result, it was harder to forecast the ablation depth than its diameter.

To more clearly illustrate the impact of the carrier density alterations on the ablation geometrical morphology, the predicted results for models considering and without considering carrier density alterations are compared in Figure 5. The two sets of data differed significantly, as expected from the results. Figure 6a compares the experimental and predicted results for ablation diameter without considering carrier density alterations. The predicted diameters were bigger when carrier density alterations were not taken into account. Ablation diameters were not correctly predicted by models without considering carrier density alterations. Figure 6b compares experimental and predicted results of the ablation depth without considering carrier density alterations. There was a notable difference when the laser power was lower than 4.35 W. It was observed from Figure 6 that the maximum predicted errors of the ablation diameter and depth were 46.30% and 73.54%, respectively.

These prediction errors might be attributed to electron-lattice energy coupling and alterations in material characteristics. Carrier density alterations affected the energy exchange between electrons and the lattice, which in turn affected the thermal conductivity and heat capacity of the material. Furthermore, the temperature distribution in the irradiated area was impacted by carrier density alterations. This study demonstrated that if the carrier density alterations were not taken into consideration in modeling, it would be challenging to determine the appropriate ablation requirement. Therefore, it was not possible to ensure the accuracy of the two-temperature model without considering carrier density alterations. In conclusion, considering the carrier density alterations in the model might enhance the prediction accuracy of the two-temperature model by providing a more accurate description of the energy exchange process in the material.

In the ablation verification tests of SiC, femtosecond laser powers of 4.35 W, 5.09 W, and 6.28 W were used for the ablation in order to confirm the guiding function of the model for the actual research. The experimental and predicted results of ablation diameter and depth are 12.1 μm, 0.35 μm and 11.2 μm, 0.37 μm, respectively, as shown in Figure 7a–c. The predicted errors of the diameter and depth of the ablation morphology were controlled within 7.44% and 5.71% at a laser power of 4.35 W, respectively. Similarly, as shown in Figure 7d–f, the experimental and predicted results of ablation diameter and depth are 14.7 μm, 0.51 μm and 14.3 μm, 0.53 μm, respectively. The predicted errors of the diameter and depth of the ablation morphology were controlled within 2.72% and 3.92% with the laser power of 5.09 W, respectively. As seen in Figure 7g–i, the experimental and predicted results of ablation diameter and depth are 17.1 μm, 0.66 μm and 17.0 μm, 0.70 μm, respectively. The predicted errors of the diameter and depth of the ablation morphology were controlled within 0.58% and 6.06% at a laser power of 6.28 W, respectively. It should be noted that the ablation depth prediction errors were substantially larger than the ablation diameter prediction errors. For all that, the established model possessed clear and effective prediction abilities.

In general, the proposed visualization model can accurately anticipate the results of single-pulse ablation. This indicates that the depth and diameter of laser-induced ablation are controllable by varying the parameters of the femtosecond laser.

### 4.2. Analysis of Ablation Geometrical Morphology under Different Parameters

The material sublimated at the high temperature caused by femtosecond laser irradiation due to the photothermal effect. Following a series of ablation trials on SiC, Figure 8 provides an overview of the variations in ablation depths and diameters under different laser powers. The results showed that as laser powers increased, ablation diameters and depths increased progressively. Nevertheless, there were not entirely linear relationships between ablation depth and laser power, as well as between ablation diameter and laser power.

Experimental results were compared to examine the impact of laser power on geometrical morphology. As shown in Figure 9, the core part of the material surface displays significant signs of ablation when exposed to a laser. In the meantime, it was evident how different laser powers produced diverse ablation morphologies. The temperature of the irradiated region increased quickly as the laser power increased. The result was that the ablation morphology gradually grew along with the increase in laser power [43].

A plasma explosion formed as a result of material ionization brought on by the femtosecond laser irradiation of SiC. However, the plasma could not be completely expelled into the air due to gravity. This portion of the plasma produced a molten material that solidified again on the ablation hole edge. Therefore, the presence of protrusions at the edge of the ablation hole can be observed in Figure 9. Because the plasma could not be completely eliminated when the laser power increased, the molten material inside the hole solidified again, resulting in poor material processing quality. Furthermore, material processing quality degraded as a result of the heat melting becoming more pronounced with increased laser power. By using cleaning or chemical corrosion techniques to get rid of surface debris that results from heat melting, surface quality might be increased.

From the experiment results, the ablation morphology could be divided into three zones: the ablation zone, the re-solidification zone, and the heat-affected zone. The ablation zone was created as a result of the material in the irradiation region instantly gasifying. Furthermore, re-solidification of the molten material occurred at the boundary of the ablation zone because the temperature was not high enough, defining the re-solidification zone. The heat-affected zone formed when the temperature in the ablation zone extended outward in all directions.

The removal mechanism during the ablation of SiC involved a complex physicochemical process that included heat conduction, energy absorption, phase explosion, plasma explosion, and other processes. Multiphoton energy was absorbed by the material during femtosecond laser irradiation, which led to ionization. The massive amount of plasma produced by the significant increase in carriers caused a plasma explosion on the material surface. Furthermore, the duration of the femtosecond laser pulse was shorter than the electron-lattice energy coupling time. Therefore, light energy was transformed into heat energy and transported into the lattice system after the multiphoton energy was absorbed by the materials. Theoretically, the ablation zone was surrounded by no significant heat-affected zones. However, a significant heat-affected zone appeared at the edge of the ablation morphology due to the effects of energy deposition and heat conduction.

To investigate the oxidation phenomenon of SiC induced by femtosecond laser, a comparison of the EDS analysis of femtosecond laser ablation with varying laser powers is presented in Figure 10. The results showed that as the laser power increased from 3.52 W to 6.28 W, the O element content reduced from 5.9% to 3.7%. This phenomenon was caused by the fact that the material removal rate increased with increasing laser intensity, which led to a decrease in the O element attached to the material. In addition, the content of the O element in the heat-affected zone is significantly higher than that in the ablation zone, as demonstrated in Figure 10. This could be attributed to the fact that femtosecond laser intensity was subject to a Gaussian distribution in space, with the energy in the outer circle of the spot being smaller [44,45].

As demonstrated by Equations (12) and (13) [46], oxides were produced when oxygen and SiC combined at high temperatures during femtosecond laser irradiation. Out of these, CO and CO_2_ evaporate during irradiation, while the silicon-oxygen compounds stick to the ablation surface. The majority of the silicon-oxygen compounds were SiO_2_. SiO_2_ was deposited on the material surface due to its compact structure. The heat effect generated by the femtosecond laser would cause the surface oxidation of SiC, which would have a substantial impact on electronic device performance. Therefore, surface oxidation was an issue that needed to be controlled in the femtosecond laser processing of SiC. The results of the experiment showed that increasing the laser intensity during the single-pulse laser ablation process resulted in a decrease in the O element content. Consequently, the surface oxidation of SiC was reduced by the efficient control of processing parameters. Higher-quality surface micro-nano structures might be processed by using a high-power femtosecond laser. However, an excessive amount of laser power might cause the molten material inside the ablation hole to solidify, which would lower the processing quality. Consequently, the appropriate selection of laser power was paramount in the context of single-pulse femtosecond laser ablation. Additionally, inert gas shielding could be used during laser processing to prevent surface oxidation. Nitrogen and argon were often utilized as inert gases in processing. Nevertheless, nitrogen would react with SiC. Consequently, femtosecond laser ablation might be carried out in an argon atmosphere to inhibit material oxidation.
(12)$${2O}_{2}(g)+\mathrm{SiC}(s)={\mathrm{SiO}}_{2}(s)+{\mathrm{CO}}_{2}\left(g\right)$$
(13)$${3O}_{2}(g)+2\mathrm{SiC}(s)={2\mathrm{SiO}}_{2}\left(s)+2\mathrm{CO}(g\right)$$

In order to investigate the impact of femtosecond laser parameters on ablation morphology in more detail, simulations with varying pulse widths and spot diameters were conducted. Figure 11 shows the predicted results for the ablation morphology of SiC. As the laser pulse width decreased, the diameter and depth of the ablation hole gradually increased. Equation (3) shows that there is a negative relationship between the laser intensity and the laser pulse width. Consequently, the greater peak intensity at shorter pulse width, which facilitates more efficient energy absorption and localized heating of the material. This study found that a shorter pulse width resulted in the formation of finer surface features. Moreover, it was clear that spot diameter affected ablation morphology. As the spot diameter increased, the ablation diameter gradually rose. The predicted results of the model only indicated that alterations in spot diameter had a little effect on the depth of the ablation hole because the depth of the ablation hole was harder to predict than its diameter. The pulse width and spot diameter were crucial in determining the final ablation outcome. The relationship between femtosecond laser parameters and ablation morphology was clarified by this work, opening the door to optimization techniques in a variety of applications, such as the processing of micro-nano structures.

In general, there is a direct relationship between the ablation morphology of SiC and femtosecond laser parameters. Analysis of ablation morphology may permit comprehension of the ablation process and be a helpful technique for precise and effective micro-nano structure formation.

### 4.3. Analysis of Carrier Density Alterations

A complicated physicochemical process including avalanche ionization, thermal conduction, plasma expansion, and laser energy absorption was responsible for the material removal caused by a femtosecond laser. In addition, the alterations in carrier density played an important role in these processes. Figure 12 shows the alterations in carrier density and laser intensity under different laser powers and pulse widths. The laser intensity and carrier density initially rose quickly, within a few hundred femtoseconds reached peaks, then dropped suddenly, remaining at zero and 0.5 × 10^27^ m^−3^, respectively. As seen in Figure 12a,c, the temporal distribution of the laser source follows a Gaussian profile. The laser pulse width and intensity had a negative relationship and the laser power and intensity had a positive relationship. In addition, Figure 12b,d illustrates alterations in carrier density followed a similar trend to alterations in laser intensity and the carrier density increased in tandem with the laser intensity. It was possible to deduce that the laser intensity had a direct impact on the alterations in carrier density when combined with the partial differential Equation (5) for carrier density. The carrier absorbed energy from the femtosecond laser and produced an interband transition, which rapidly raised the temperature and increased the number of free carriers.

According to recent studies, SiC was irradiated by a femtosecond laser, which caused multiphoton ionization and avalanche ionization to create massive surface plasmas [47]. Many free carriers were created when the laser pulse created plasmas on the material surface. The density of free carriers was a crucial feature of the initial plasmas, which significantly impacted the geometrical morphology of SiC. Carrier density alterations had an impact on the overall energy transfer and heat accumulation in the material. Furthermore, the absorption and reflection of laser energy were determined by the carrier density, which had a direct impact on the material-laser interaction process. The higher carrier density could lead to increased energy absorption and localized heating, which would expand the ablation morphology. Therefore, comprehending the alterations in carrier density during femtosecond laser processing provided important theoretical support for manipulating ablation morphology.

Figure 13 shows the alterations in absorption coefficient by free carriers and reflectivity under different laser powers and pulse widths. When a femtosecond laser was used to irradiate a material, a significant number of free carriers were produced. This caused the material surface to respond transiently in the free carrier absorption coefficient and reflectivity. The peak value of the free carrier absorption coefficient increased as laser power increased and pulse width decreased, as seen in Figure 13a,c. This trend was consistent with the alterations in carrier density. The production of plasmas and subsequent ionization of free carriers during femtosecond laser irradiation had a major effect on the free carrier absorption coefficient of SiC. The production of numerous carriers during laser irradiation led to an increase in the absorption coefficient of free carriers. At this stage, SiC was absorbing a significant quantity of energy and the temperature was rapidly rising.

From Figure 13b, it can be seen that the reflectivity remained unchanged during the first 300 fs and improved to its maximum value at 1500 fs. As shown in Figure 13d, pulse width can not only change the rate and peak of reflectivity raised but also affect the initial time when reflectivity remained unchanged. The alterations in reflectivity caused corresponding alterations in carrier density. The material absorbed a large amount of energy since its initial reflectivity was almost zero, which led to a notable increase in carrier density. After 500 fs, the carrier density started to decline as reflectivity quickly increased and the energy absorbed by the SiC was significantly decreased. At 1500 fs, the phenomenon of plasma shielding occurred [48,49], causing the carrier density to approach 0.5 × 10^27^ m^−3^.

The alterations in heat capacity and thermal conductivity of electrons and lattices were studied to reveal the intricate balance between heat capacity and carrier density, as well as between thermal conductivity and carrier density. It is evident from Equations (1) and (2) that temperature plays a pivotal role in heat capacity and thermal conductivity alterations. These alterations were graphically depicted in Figure 14 for electron thermal conductivity and heat capacity under varying conditions. It might be assumed that alterations in carrier density caused alterations in thermal conductivity and heat capacity because of the impact of carrier density on temperature. Figure 15 displays the alterations in lattice thermal conductivity and heat capacity under various conditions. In addition to affecting electron heat capacity and thermal conductivity, laser power and pulse width also had an effect on lattice heat capacity and thermal conductivity.

The electron and lattice heat capacities are comparatively small, as seen in Figure 14 and Figure 15. Consequently, the material heating caused by the laser occurs extremely quickly. Crucially, the lattice and electron temperature alterations were two separate processes. The two-temperature model demonstrated that the carrier density alterations affected the heat sources of the lattice system and led to a redistribution of the lattice temperature.

In general, the energy coupling between the femtosecond laser and SiC is significantly influenced by the carrier density. As the laser intensity increased, many free carriers were produced. Carrier density directly affects the electron-lattice coupling coefficient and induces lattice temperature redistribution. Therefore, taking into account the carrier density alterations enhances the predictive accuracy of the model and allows for fine control of ablation morphology.

## 5. Conclusions

To study the ablation morphology of SiC, this work suggested a multi-physics modeling that considered carrier density alterations. Relevant experiments were conducted in which SiC was ablated using various femtosecond laser powers. To validate the models, the outcomes of the simulation were compared with the experimental data. This work offered a novel approach to the parameter selection of femtosecond laser. This study results may include the following:
(1)The experimental results verified that the model could accurately predict the ablation morphology considering alterations in carrier density during the ablation of SiC. The predicted errors of ablation diameter and depth within a specific range were controlled within 7.44% and 6.94%, respectively. The multi-physics field model developed in this work can be used to establish the quantitative relationship between the ablation morphology and the laser power.(2)The removal mechanism during the ablation of SiC was analyzed. Multiphoton absorption is thought to be the primary energy deposition mechanism in femtosecond ablation. The deposition of laser pulses and heat conduction induce a very strong thermal effect on the material surface.(3)The response relationship analysis of the nonlinear alteration in optical as well as physical parameters and the ablation morphology was developed. The carrier density alterations directly affect the electron-lattice coupling coefficient, resulting in lattice temperature redistribution. As the carrier density increases, the temperature rises rapidly, leading to an increase in the ablation depth and diameter.


## Figures and Tables

**Figure 1 micromachines-15-00573-f001:**
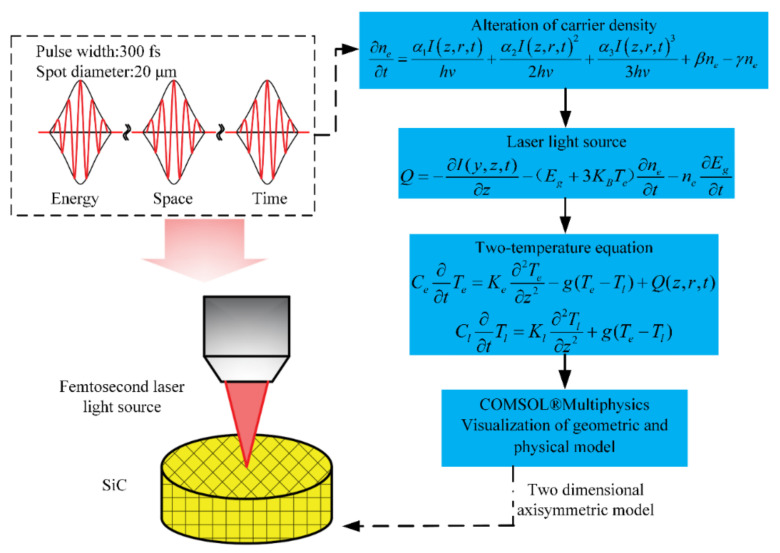
Modeling process flowchart.

**Figure 2 micromachines-15-00573-f002:**
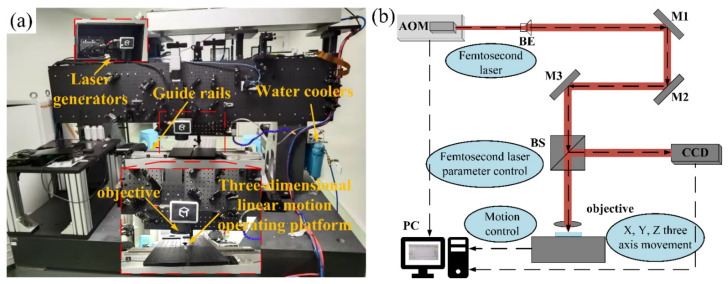
The femtosecond laser processing setup: (**a**) experimental setup and (**b**) the schematic diagram.

**Figure 3 micromachines-15-00573-f003:**
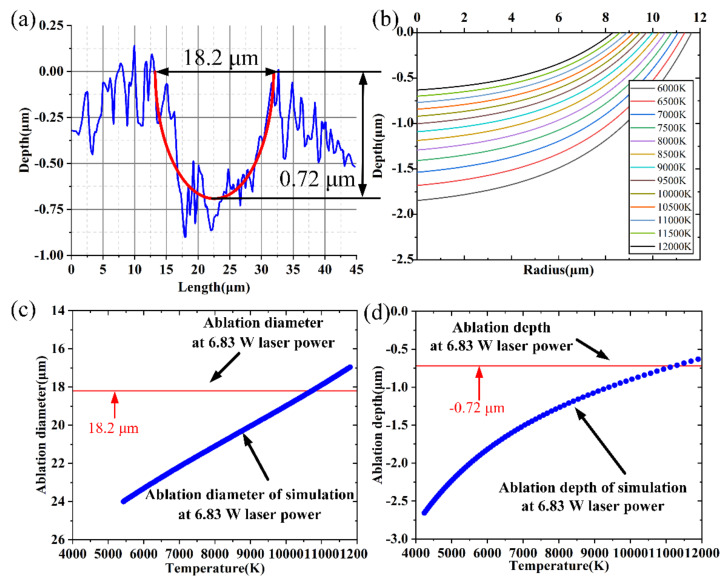
Experimental and predicted results at 6.83 W laser power: (**a**) CLSM image of SiC, (**b**) isotherm distribution at various temperatures, (**c**) ablation diameter, (**d**) ablation depth.

**Figure 4 micromachines-15-00573-f004:**
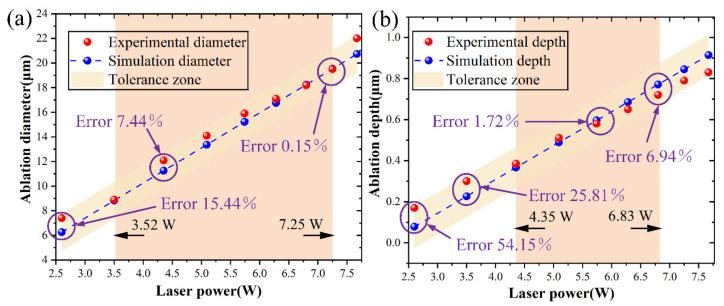
Experimental and predicted results: (**a**) ablation diameter and (**b**) ablation depth.

**Figure 5 micromachines-15-00573-f005:**
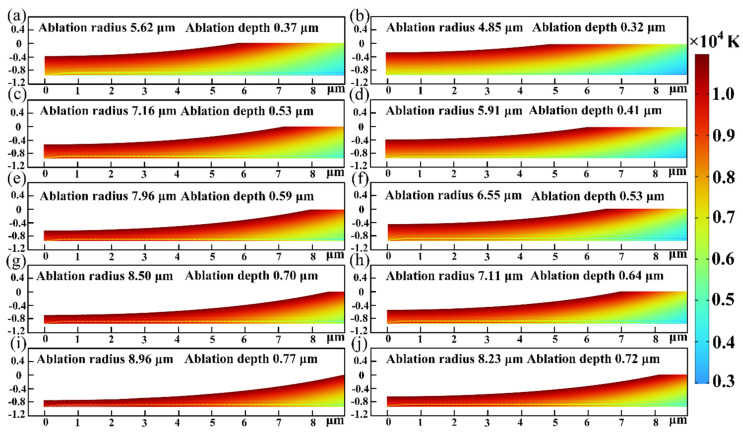
Predicted results of ablation morphology under different femtosecond laser powers (the left column considering carrier density alterations, the right column without considering carrier density alterations): (**a**,**b**) 4.35 W, (**c**,**d**) 5.09 W, (**e**,**f**) 5.74 W, (**g**,**h**) 6.28 W, and (**i**,**j**) 6.83 W.

**Figure 6 micromachines-15-00573-f006:**
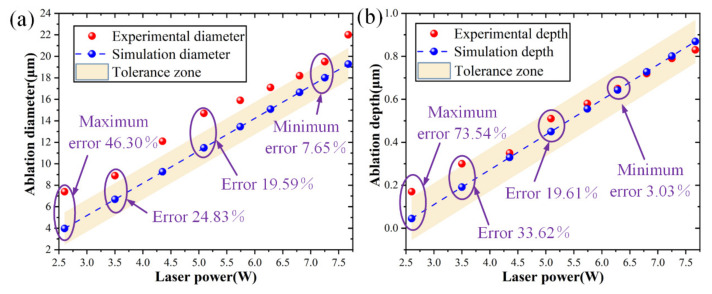
Experimental and predicted results of the ablation morphology without considering carrier density alterations: (**a**) ablation diameter and (**b**) ablation depth.

**Figure 7 micromachines-15-00573-f007:**
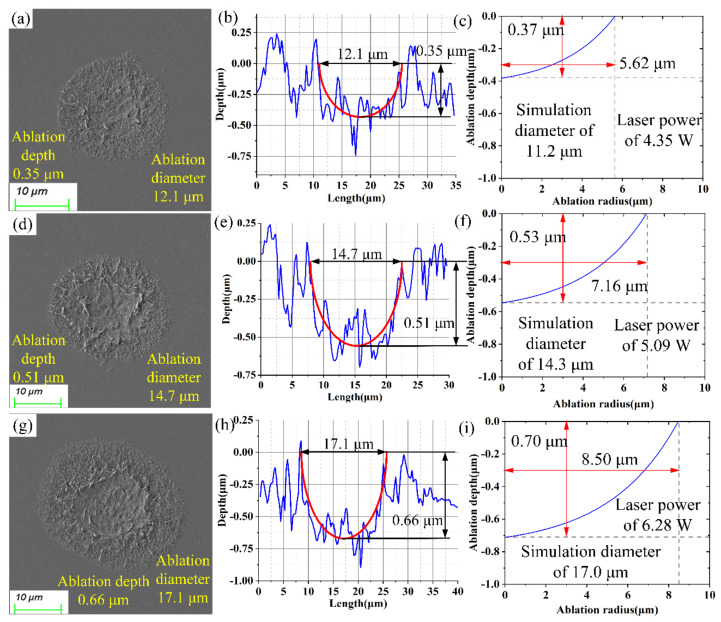
Experimental and predicted results of the ablation morphology: (**a**,**d**,**g**) SEM images at laser powers of 4.35 W, 5.09 W, and 6.28 W, (**b**,**e**,**h**) CLSM images at laser powers of 4.35 W, 5.09 W, and 6.28 W, (**c**,**f**,**i**) predicted images at laser powers of 4.35 W, 5.09 W, and 6.28 W.

**Figure 8 micromachines-15-00573-f008:**
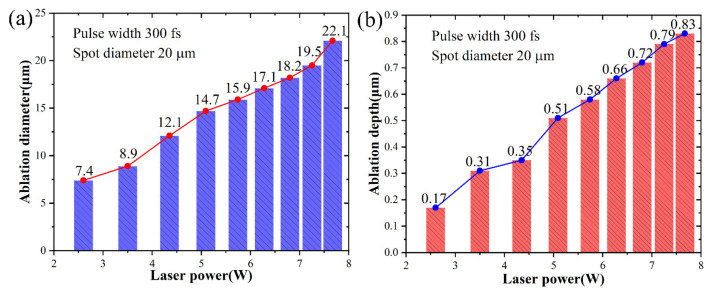
The experimental results of ablation diameter and depth at different laser powers: (**a**) ablation diameter and (**b**) ablation depth.

**Figure 9 micromachines-15-00573-f009:**
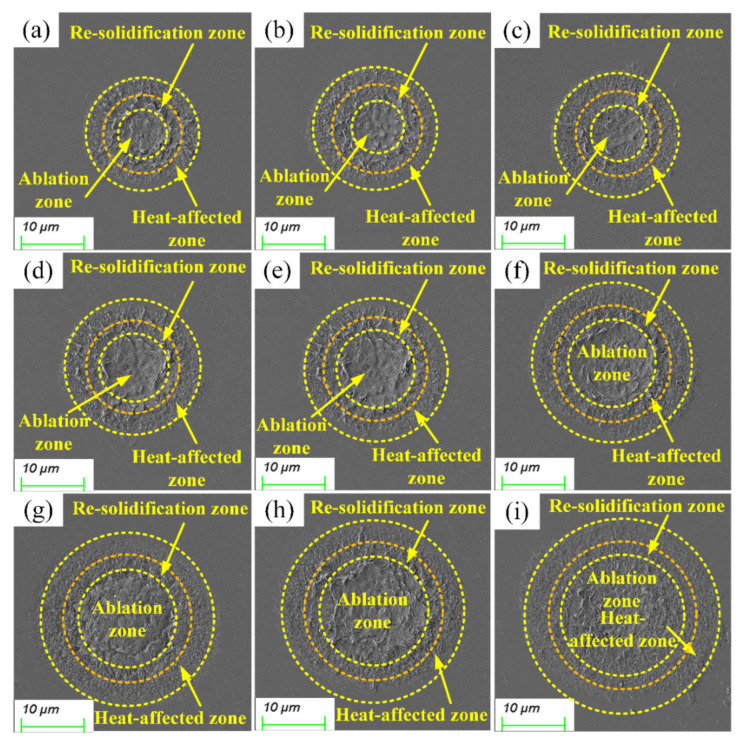
The SEM micrographs produced by femtosecond laser radiation at various laser powers: (**a**) 2.64 W, (**b**) 3.52 W, (**c**) 4.35 W, (**d**) 5.09 W, (**e**) 5.74 W, (**f**) 6.28 W, (**g**) 6.83 W, (**h**) 7.25 W, and (**i**) 7.67 W.

**Figure 10 micromachines-15-00573-f010:**
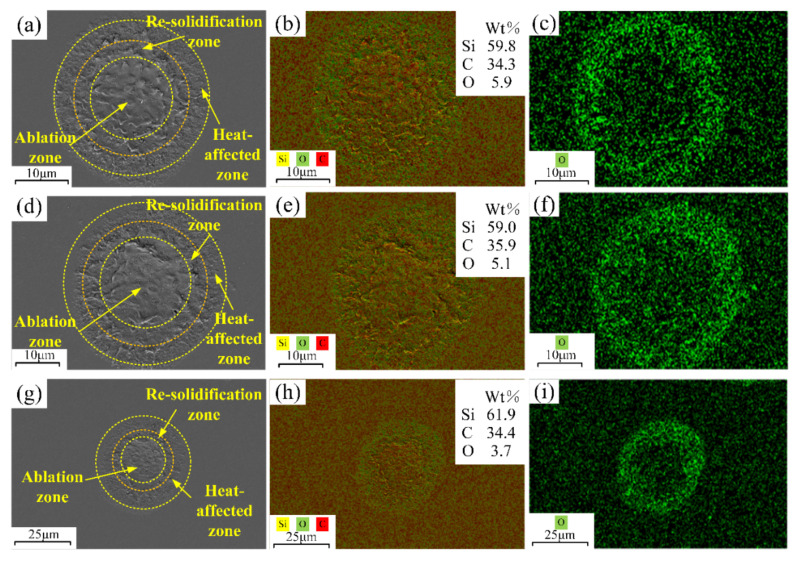
The EDS analysis for SiC ablation using femtosecond laser at varying powers: (**a**–**c**) 3.52 W, (**d**–**f**) 5.09 W and (**g**–**i**) 6.28 W.

**Figure 11 micromachines-15-00573-f011:**
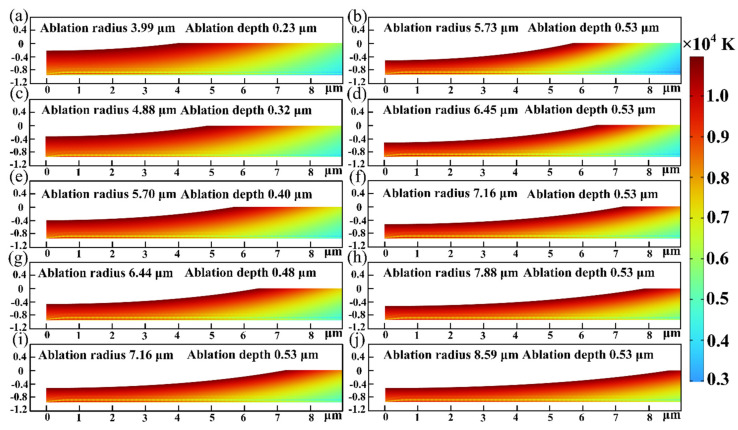
Predicted results of ablation morphology under different parameters: (**a**,**c**,**e**,**g**,**i**) the spot diameter of 20 μm and pulse widths of 500 fs, 450 fs, 400 fs, 350 fs, 300 fs and (**b**,**d**,**f**,**h**,**j**) the pulse width of 300 fs and spot diameters of 16 μm, 18 μm, 20 μm, 22 μm, 24 μm.

**Figure 12 micromachines-15-00573-f012:**
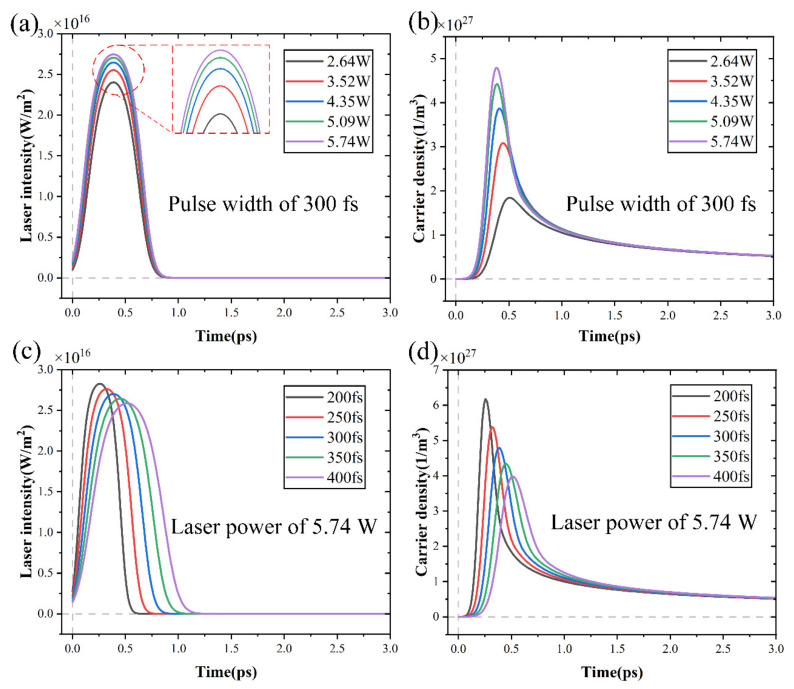
The alterations in laser intensity and carrier density with time: (**a**,**b**) under different laser powers, (**c**,**d**) under different pulse widths.

**Figure 13 micromachines-15-00573-f013:**
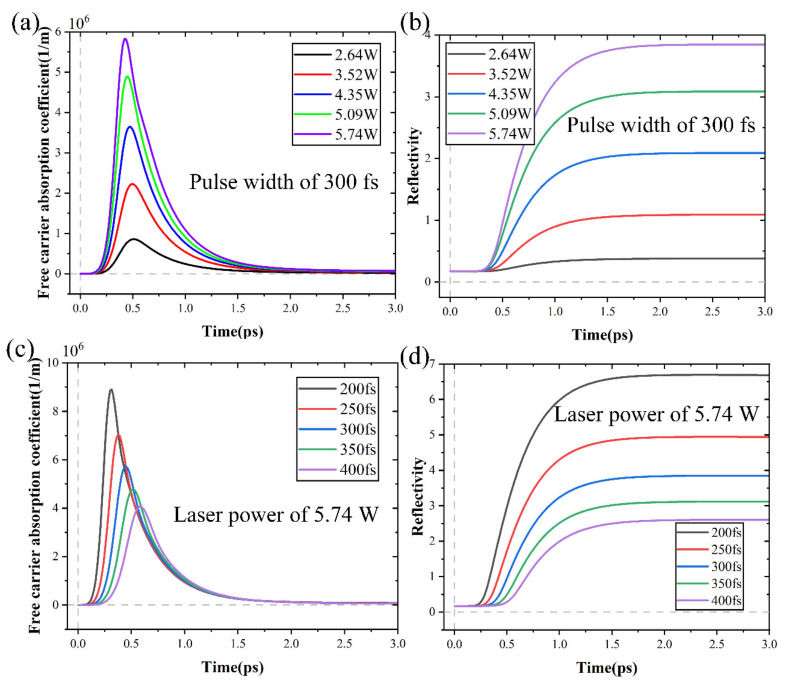
The alterations in absorption coefficient by free carriers and reflectivity with time: (**a**,**b**) under different laser powers, (**c**,**d**) under different pulse widths.

**Figure 14 micromachines-15-00573-f014:**
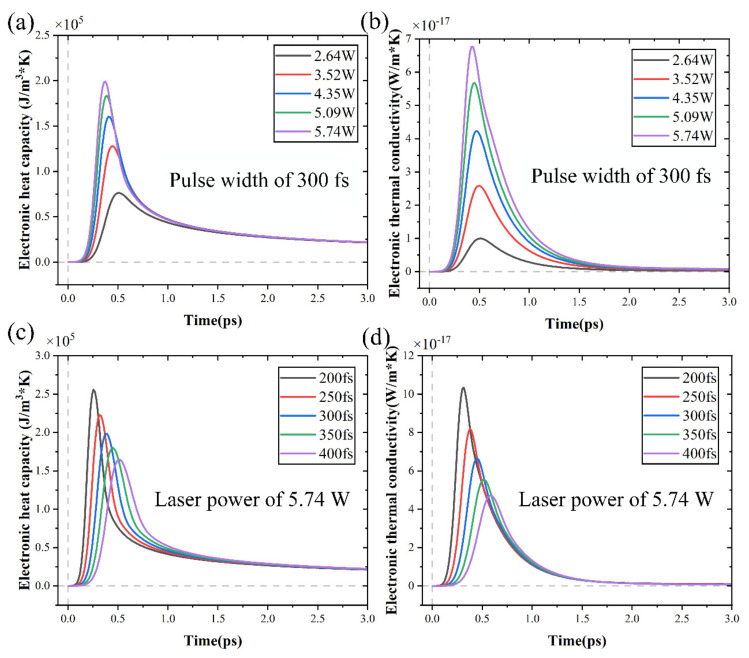
The alterations in heat capacity and thermal conductivity of electron with time: (**a**,**b**) under different laser powers, (**c**,**d**) under different pulse widths.

**Figure 15 micromachines-15-00573-f015:**
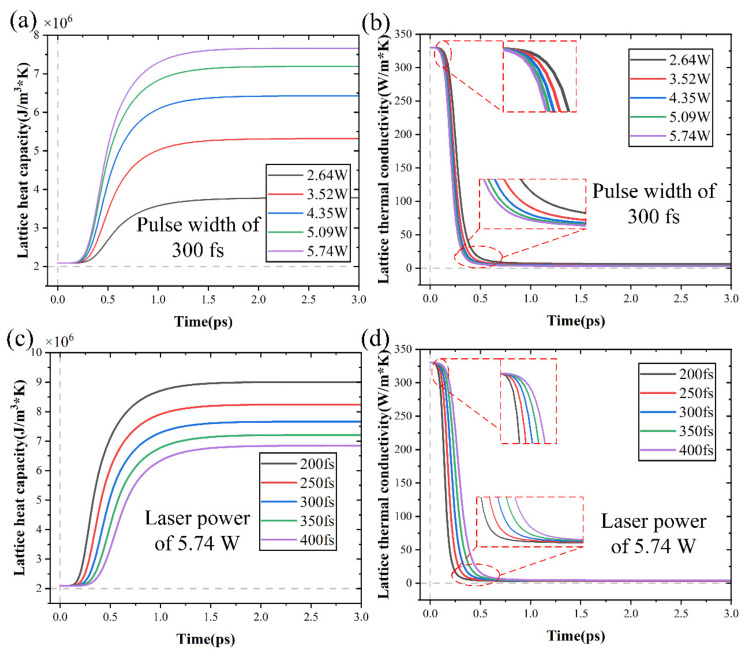
The alterations in heat capacity and thermal conductivity of lattice with time: (**a**,**b**) under different laser powers, (**c**,**d**) under different pulse widths.

**Table 1 micromachines-15-00573-t001:** The laser parameters.

Parameters	Values
Maximum power	7.67 W
Power stability	0.12%
Laser wavelength	1030 nm
Pulse width	300 fs
Temperature	21 °C

## Data Availability

The data presented in this study are available on request from the corresponding author. The data are not publicly available due to the need for further research.

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
