# Peer review of "Experimental and Simulation Research on Femtosecond Laser Induced Controllable Morphology of Monocrystalline SiC"

_micromachines, 2024, doi:10.3390/mi15050573_

Round 1
Reviewer 1 Report
Comments and Suggestions for Authors
Referee report
The article “Experimental and simulation research on femtosecond laser induced controllable morphology of monocrystalline SiC” is dedicated to quite actual topic - the pulse laser modification of materials. Suffice it to say, that this year's Nobel Prize in Physics was awarded for extreme-ultrashort pulses of electromagnetic radiation. Pulsed laser crystallization is a very efficient approach. The article contains new experimental data and will be interesting for researchers and technologists. It is especially interesting and useful for practical use that the authors modify quite wide-gap material (SiC) using femtosecond laser pulses. The article can be published after minor revision.
Comment:
The authors indicate the laser power and spot area. However, the laser energy density per pulse, the so-called fluence, is very important. So, it is important what shape did the laser spot have - Gaussian? As for laser fluence, it should be clarified, is this the fluence at the center of the Gaussian beam? Authors should briefly calculate fluence, such as in the articles Yuzhu Cheng, Alexander V. Bulgakov, Nadezhda M. Bulgakova, Jiˇrí Beránek, Martin Zukerstein, Ilya A. Milekhin, Alexander A. Popov and Vladimir A. Volodin. Ultrafast Infrared Laser Crystallization of Amorphous Ge Films on Glass Substrates. MDPI Micromachines 14, 2048 (2023). DOI: https://doi.org/10.3390/mi14112048 or “Single-shot selective femtosecond and picosecond infrared laser crystallization of an amorphous Ge/Si multilayer stack” Optics and Laser Technology, v.161, 109161 (2023) DOI: https://doi.org/10.1016/j.optlastec.2023.109161 and references therein.
Accept with minor revision.
Reviewer 2 Report
Comments and Suggestions for Authors
It is known that SiC is an ideal material for applications in power electronics, light-emitting diodes, sensors and aero engines. Special surface micro- and nano-structures can enhance the mechanical, optical and electrical properties of SiC. Femtosecond laser has been proved to be an effective way for designing and processing various micro- and nano-structures on SiC. Recently, there have been many reports on the femtosecond laser ablation behavior of SiC, including the femtosecond laser induced oxidation behavior, structural transformation, thermal effects. From the point of view, it is worthy in-depth study in the exploration of SiC micro- and nano-structures processing methods.
The authors report an interesting and significant work on the femtosecond laser machining of SiC. Therefore, this paper can be accepted after minor revision. The comments are as follows:
1. What is the reason to choose the specific laser powers( 2.64W, 3.52W, 4.35W, 5.09W, 5.74W, 6.28W, 6.83W, 7.25W, 7.67W) in this work?
2. The author should explain if and how the reported study adds novel information or not?
3. The conclusions are still a list of observations. Please advance the understanding to separate the study from a technical report from a scientific article.
4. The removal mechanism and oxidation behavior should be discussed in detail.
Comments on the Quality of English LanguageA spelling and grammar check would also be most needed.
Reviewer 3 Report
Comments and Suggestions for Authors
In their paper, the authors propose a study about the ablation of Silicon Carbide, which is a material difficult to precisely ablate due to its hardness. By performing an analysis based on both physical simulations and experimental tests, they show to be able to accurately predict the ablation properties depending on the irradiation parameters.
The paper is experimentally solid, clear and well presented, and it could be of interest to the material processing community. Therefore, I suggest its publication after a correction of the English language (see suggestions in the dedicated area)
Comments on the Quality of English LanguageIn both the abstract and the introduction, the past tense is used for no reason even where a present tense would be more appropriate. This should be corrected, because it makes the paper more difficult to read
Reviewer 4 Report
Comments and Suggestions for Authors
The paper “Experimental and simulation research on femtosecond laser induced controllable morphology of monocrystalline SiC” by Yang Hua et al. illustrates both the physical model and simulations, as well as numerous experimental results, indicating a thorough investigation of SiC processing with fs laser. In general, the manuscript is written clearly, and its structure is well-organized. However, I have a few suggestions to enhance the clarity of the paper.
Practically the entire manuscript is written in the past tense, whereas the present tense could be used in many parts.
L32: “wide bandgap”
L47: What do the authors mean by 'times'?
L91: please, add the subject
L123: “traditional long-pulsed lasers”: The term “long-pulsed” doesn't seem appropriate
L131: “that is”
L214: The term “remove” doesn't seem appropriate
Fig. 5: please, check the caption, as well as its description in the text. It is not clear the color scale.
L329-330: “surface displayed significant ablation morphology” is not clear
L370: “issue” or “undesirable outcome” would be preferable instead of “defect”
Fig. 11: insert a comment of the color scale.
L417-418: “The laser pulse width and intensity had a negative correlation and the laser power and intensity had a positive correlation.” The term ‘correlation’ does not seem appropriate.
Finally, in the introduction, the authors should also mention the applications of laser-treated Sic (see for example, https://doi.org/10.1007/978-94-011-3842-0_2, https://doi.org/10.3390/app10207095, https://doi.org/10.1016/j.sna.2018.04.029).
Based on the aforementioned comments, I suggest a minor revision of the paper.
Comments on the Quality of English Language
see "Suggestions"
